# Acute Effects of *Hibiscus sabdariffa* Calyces on Postprandial Blood Pressure, Vascular Function, Blood Lipids, Biomarkers of Insulin Resistance and Inflammation in Humans

**DOI:** 10.3390/nu11020341

**Published:** 2019-02-05

**Authors:** Salisu M. Abubakar, Moses T. Ukeyima, Jeremy P. E. Spencer, Julie A. Lovegrove

**Affiliations:** 1Hugh Sinclair Unit of Human Nutrition, Department of Food and Nutritional Sciences and Institute for Cardiovascular and Metabolic Research, University of Reading, Reading RG6 6AP, UK; smabubakar.bch@buk.edu.ng (S.M.A.); mosesukeyima@gmail.com (M.T.U.); j.p.e.spencer@reading.ac.uk (J.P.E.S.); 2Department of Biochemistry, Bayero University, P.M.B. 3011 Kano, Nigeria; 3Department of Food Science and Technology, University of Agriculture, P.M.B. 2373 Makurdi, Nigeria; 4Molecular Nutrition Group, Centre for Integrative Neuroscience and Neurodynamics, University of Reading, Reading RG6 6AP, UK

**Keywords:** *Hibiscus sabdariffa* calyces, postprandial blood pressure, vascular function

## Abstract

Background/Objectives: The acute impact of *Hibiscus sabdariffa* calyces (HSC) extract on postprandial vascular function and other cardiometabolic risk factors have not been studied previously. This study investigated the acute impact of HSC extract consumption on blood pressure (BP), vascular function and other cardiometabolic risk markers. Subjects/Methods: Twenty-five men with 1% to 10% cardiovascular disease (CVD) risk (determined by QRISK^®^2) were randomised to consume either 250 mL of the aqueous extract of HSC or water with breakfast in a randomised, controlled, single-blinded, 2-meal cross-over study (ClinicalTrials.gov, NTC02165553) with a two weeks washout period between study days. BP was measured at baseline and hourly for 4 h. Flow mediated dilatation (FMD) of the branchial artery was measured at baseline, 2 and 4 h post intervention drink consumption. Results: Acute consumption of aqueous extract of HSC caused a significant increase in % FMD (*p* < 0.001), a non-significant decrease in systolic BP (SBP) and diastolic BP (DBP); non-significant increase in urinary and plasma nitric oxide (NOx) and reduced response of serum glucose, plasma insulin, serum triacylglycerol and C-reactive protein (CRP) levels; significant (*p* = 0.026) improvement in the area under systemic antioxidant response curve (0 to 2 h); no significant changes in arterial stiffness following the acute consumption of the extract of HSC. Gallic acid, 4-*O*-methylgallic acid, 3-*O*-methylgallic acid and hippuric acid reached a maximum plasma concentration at 1 to 2 h post consumption of the extract of HSC. Conclusion: The extract of HSC improved postprandial vascular function and may be a useful dietary strategy to reduce endothelial dysfunction and CVD risk, although this requires confirmation.

## 1. Introduction

Cardiovascular diseases (CVD) are the leading cause of death worldwide and have become a global public health concern. CVD are the main cause of death in developed [1,2] and developing countries [3]. According to the World Health Organisation (WHO), CVD cause 17.5 million deaths in the world each year [4,5]. It is projected that annual death due to CVD will increase from 17 million in 2008 to 25 million in 2030 [5]. Hypertension is the first and most important modifiable risk factors for CVD. In addition, markers of vascular endothelial function are regarded as novel or emerging CVD risk markers. Measurement of flow mediated dilatation (FMD) of the brachial artery is regarded as the gold standard measure of vascular endothelial function. Dietary modification is believed to be a major contributor to CVD risk reduction. Foods and beverages rich in polyphenols are thought to have a potential positive impact on CVD risk.

There are over 300 species of *Malvaceae*, and *Hibiscus sabdariffa* is the most commonly known. Infusions of *Hibiscus sabdariffa* calyces (HSC) are consumed in most parts of the world as a cold beverage or a hot drink and are believed to have antihypertensive and hypolipidaemic activities [6,7]. HSC are polyphenol-rich and are considered by some to have nutraceutical potential which is under exploited [8]. It has been postulated that the hypotensive effect of the extract of HSC could be associated with a number of potential mechanisms, including direct vasorelaxant effects [9,10]. The hypotensive effect of the extract of HSC was proposed to be related to its ability to induce endothelium-dependant effects related to the nitric oxide synthase (NOS) activation by active components within the extract. In addition, the extract of HSC has been found to act as an inhibitor of angiotensin converting enzyme (ACE) in vitro, [11,12]. ACE catalyses the conversion of angiotensin I to angiotensin II (vasoconstrictor) leading to an increase in blood pressure (BP). Angiotensin II promotes an increase in extracellular fluid volume through the increased sodium and water intake and retention coupled with a decrease in excretion of sodium and water, which results in a BP increase. ACE inhibitors are potent hypotensive agents. HSC anthocyanins (delphinidine-3-*O*-sambubioside and cyanidine-3-*O*-sambubioside) were reported to competitively inhibit ACE from rabbit lung [12]. This mechanism of action is supported in human studies which related hypotensive activity of the extract of HSC to inhibition of ACE and decrease serum sodium [13]. The polyphenols in the extract of HSC were thought to act through multi-faceted metabolic regulation [14].

Biochemical markers of dyslipidaemia, insulin resistance/sensitivity, inflammation and oxidative stress are among the modifiable cardiovascular risk markers shown to be responsive to dietary (fruits and vegetables) intervention, but evidence from human intervention trials appears to be limited as most studies called for further research [15,16,17,18,19,20,21,22,23,24,25,26].

The limited human studies on the impact of hibiscus drink consumption on cardiometabolic risk markers were of chronic design. The biological activities of phenolic compounds in the promotion of health and disease prevention depend on the intake, absorption, transport to target organs and metabolism [27]. Despite claims that the benefits of the consumption of the extract of HSC to cardiovascular health are related to its polyphenols content, only one study has investigated the pharmacokinetics of HSC anthocyanins in humans to date [28]. In addition, there have been no human studies that investigate the acute impact of the consumption of the extract of HSC on vascular function, arterial stiffness and nitric oxide. 

Based on the limited data and suggestions for more clinical trials to confirm the benefits of the consumption of HSC extract, this study investigated the acute impact of the consumption of the aqueous extract of HSC on postprandial FMD, BP and other cardiometabolic risk markers in men.

## 2. Subjects and Methods

### 2.1. Ethical Approval

The study protocol was reviewed and given a favourable ethical opinion for conduct by the University of Reading Research Ethics Committee (UREC) (Project No. 13/40). The study protocol was registered at the ClinicalTrials.gov (NTC02165553). The study was conducted in accordance with the Declaration of Helsinki [29,30] and adhered to the Consolidated Standards of Reporting Trials (CONSORT) [31], as shown in Figure 1.

### 2.2. Study Design

This study was a randomised, controlled, single-blinded, acute, cross-over trial conducted in the Hugh Sinclair Unit of Human Nutrition, University of Reading. The study consisted of two dietary intervention arms: polyphenol-rich hibiscus drink and control drink (water) and investigated the impact of their acute consumption on CVD risk markers in human volunteers with 1 to 10% CVD risk in ten years (measured by QRISK^®^2 see below). Volunteers were randomised using a computer-based randomisation software (Research Randomizer, Geoffrey C. Urbaniak and Scott Plous) and were assigned to consume a glass (250 mL) of hibiscus or water with a high fat meal, at time 0 min followed by a medium fat lunch at 120 min in a random order separated by a two-week washout period.

The study breakfast and lunch were prepared in the Hugh Sinclair Unit of Human Nutrition, University of Reading, UK. The breakfast consisted of 120 g buttered croissants, 30 g butter, a spoonful (15 g) of clear honey and 250 mL of hibiscus drink or low nitrate mineral water (Buxton, Nestle Waters UK Ltd, Gatwick, West Sussex, UK). The lunch consisted of 2 slices (89 g) of thick soft white bread, 33 g of soft cheese, 2 shortbread biscuit fingers (35 g), a bag (18 g) of ready salted crisps and Buxton mineral water. A summary of nutrients in the study meals is shown in Table 1.

### 2.3. Sample Size and Power Calculation

This study considered systolic BP (SBP) as the primary outcome measure. Previous chronic study findings [6] showed an average of 12.7 mmHg reduction of SBP with a standard deviation of 6.8 to 11.8 mmHg after consumption of Hibiscus drink. On this basis, and assuming two-sided test criteria with alpha of 0.05, mean SBP difference of 5 mmHg and standard deviation of 6 mmHg for an acute study, a sample size of 23 was required for a statistical power of 80%. The calculation was performed as previously described [32,33]. Twenty-five (25) participants were recruited to account for a 10% drop-out.

### 2.4. Subjects Enrolment

The study was advertised within and around the University of Reading, Reading, UK via posters, emails to volunteers on the Hugh Sinclair volunteer database (who consented to be contacted for potential participation on intervention trials), targeted social media and word of mouth. Volunteers who declared interest were sent a detailed study information sheet and were required to fill and return a health and lifestyle questionnaire to assess their eligibility for screening. 

The inclusion criteria were that volunteers should be male, have 1 to 10% CVD (QRISK^®^2) risk in 10 years, not taking BP medication, not having kidney, liver or chronic diseases, smoking or not smoking and signed informed consent. Those who were eligible for screening were invited to the Hugh Sinclair Unit of Human Nutrition and requested to give informed written consent after any questions were answered. This was followed by BP and anthropometric measurement collection, and a screening fasted blood sample was collected and tested for glucose, lipids, liver and renal function markers and full blood count. Volunteers who were found eligible for the study were informed and with the aid of a computer-based randomisation software randomly assigned to an intervention order.

### 2.5. CVD Risk Calculation

An online version of QRISK^®^2 CVD risk calculator recommended by the National Institute for Health and Care Excellence (NICE) was used to estimate the CVD risk of volunteers [34]. 

### 2.6. Hibiscus Drink Preparation

Hibiscus drink was prepared in the Hugh Sinclair Unit of Human Nutrition kitchen in batches on a weekly basis. A batch was prepared in comparison with the usual method employed by habitual tea consumers. Briefly, a total of 30 g (average of 15 hibiscus tea bags) of *H. sabdariffa* calyces were steeped in 1 litre of low nitrate water (Buxton mineral water, nitrate content <0.1 mg/L) at 100 °C for 10 min. The tea bags were squeezed to ensure maximum extraction of the bioactive. The tea was allowed to cool at room temperature and a 250 mL aliquot was added to an opaque, food grade aluminium bottle, cap labelled and refrigerated at 4 °C until required for use. Each volunteer was served an equivalent extract of 7.5 g *Hibiscus sabdariffa* calyces in 250 mL Buxton water. Each serving contained a mean of 150 mg total anthocyanins and 311 mg gallic acid.

### 2.7. Blood and Urine Samples Collection and Processing

Volunteers arrived at the Hugh Sinclair Unit of Human Nutrition fasted having drunk only low-nitrate mineral water (Buxton mineral water) for the past 12 h and abstained from polyphenol-rich foods 24 h prior to the study visit. A flexible cannula was inserted into the antecubital vein of their left arm to enable regular blood sampling. Blood samples were collected into BD K3-EDTA, heparin and serum separation vacuette (Greiner Bio-One, Monroe, Louisiana, USA). EDTA and heparinised samples were immediately centrifuged at 4 °C and 3000 rpm for 10 min. Blood in serum separation tubes (SST) were allowed to stand at room temperature for 30 min before centrifugation. Plasma and serum samples were aliquoted and stored at −20 °C until future analysis.

Urine samples were collected into calibrated plastic containers and the volume measured at each point. The samples were mixed, aliquot into centrifuge tubes and centrifuge at 4 °C and 3000 rpm for 10 min; the supernatant was collected into labelled sample tubes and stored at −20 °C until required for analysis. 

### 2.8. Measurement of FMD of the Brachial Artery

FMD of the brachial artery was measured at baseline (0), 2, and 4 h after consumption of the high fat breakfast with either hibiscus drink or water. FMD of the brachial artery was measured in accordance with the guidelines in the Report of the International Brachial Artery Reactivity Task Force [35] and previous descriptions [36,37,38,39,40]. FMD measurements were captured using HDI^®^ 5000 ALT Ultrasound System (Philips Medical Systems, Nashville, Tennessee, USA) connected with a 5–10 MHz hockey-stick probe with a dimension of 3.5 × 1.0 cm. Brachial artery images captured by the ultrasound machine were analysed using brachial analyser software (Medical Applications, LLC, Coralville, Iowa, USA) on the computer connected to the ultrasound. The diameter of the branchial artery throughout the length of the lumen was captured and used for analysis.

Brachial artery FMD was calculated as the percentage change in artery diameter relative to baseline as in the following equation:

% FMD = [PAD (mm) − BAD (mm)/BAD (mm)] × 100

where; PAD is Peak Artery Diameter = maximum artery diameter obtained at post-occlusion BAD is the Baseline Artery Diameter = average of baseline (pre-occlusion) artery diameters.

### 2.9. Assessment of Arterial Stiffness

The central BP and indices of arterial stiffness were measured using radial artery pulse wave analysis (SphygmoCor^®^ software version 9.0, AtCor Medical Pty Ltd., Sydney, Australia) as described in the manufacturer’s manual and published previously [38,39] and following recommendations of the experts’ consensus on arterial stiffness measurement methods [41] 

### 2.10. BP Measurement

On arrival at the Hugh Sinclair Unit of Human Nutrition volunteers were requested to relax in bed in an air-conditioned (23 °C) clinical room for 30 min before BP measurement. BP was measured using automatic BP monitor (OMRON Healthcare Co. Ltd., Kuoto, Japan) at baseline (0) and 1, 2, 3 and 4 h after consumption of a high fat breakfast with either hibiscus drink or water. The mean of three consecutive readings taken on the right arm within 10 min was taken at each measurement point. The heart rate was also determined. Pulse pressure (PP) was measured as the difference between SBP and DBP [42,43].

### 2.11. Solid Phase Extraction (SPE) and HPLC Analysis of Plasma Polyphenols 

The preparation of plasma samples for analysis of anthocyanins, phenolic acids and their metabolites was performed as previously described [38,44,45] with modifications. In brief, plasma (500 µL) samples were diluted by addition of 1 mL of 0.5% glacial acetic acid. To this, 50 µL of 100 µM 3-(4-Hydroxy-3-methoxyphenylpropionic acid), 3hmPA, was added as a recovery standard. The mixture was vortexed and centrifuged at 17,000× *g*, 4 °C for 15 min. The supernatant was loaded onto an Oasis HLB 60 mg 3 cc cartridge (Waters Limited, Elstree, Hertfordshire, UK) earlier conditioned by sequential addition of 1 mL each of acidified methanol (0.1% Trifluoroacetic acid, TFA, pH 2, in methanol) and acidified water (0.5% glacial acetic acid in distilled water). The SPE cartridge was then washed by sequential addition of 3 mL acidified water (0.5% glacial acetic acid in distilled water) and 1 mL of 0.5% acidified water in 20% methanol (water:methanol:acetic acid, 79.5:20:0.5 respectively). The cartridge was then dried by applying a vacuum for 5 to 10 s. This was followed by elution of phenolics, into speedvac tubes containing 200 µL of acidified methanol (0.5% acetic acid in methanol), by adding 1 mL of acidified methanol (0.1% Trifluoroacetic acid, pH 2, in methanol) onto the solid phase extraction cartridge twice, and passing through the SPE cartridge gravitationally. Elusion was completed with the application of a vacuum for 5 to 10 s to maximise recovery.

The eluent was dried using a speedvac system (Thermo Fisher Scientific Inc. Basingstoke, Hampshire, UK) and redissolved with 200 µL of HPLC mobile phase A (0.1% formic acid in HPLC water). The redissolved eluent was then filtered through a 0.20 µm filter (Sartorius Stedim Biotech, Gottingen, Germany) into HPLC vial and ran on an Agilent 1100 series (Agilent Technologies Ltd. Cheadle, Manchester, UK). The HPLC system is equipped with a diode array detector and Nova-Pak C_18_ column (250 × 4.6 mm; 4 µm particle size; 30 °C; Waters Limited, Elstree, Hertfordshire, UK). The mobile phase A is 0.1 % formic acid in HPLC water while mobile phase B contained 0.1% HPLC water in acetonitrile and were pumped through the column at 0.3 mL per minute. Samples (50 µL) were injected and separated using the following gradient system (min/% mobile phase A/%mobile phase B): 0/95/5, 1/95/5, 20/70/30, 25/20/80, 30/20/80, 31/95/5 and 45/95/5. The eluent was monitored by photodiode array detection at 280 nm (phenolic acids and their metabolites) and 520 nm (anthocyanins and their metabolites) with spectra of products obtained between 220–600 nm. Retention time and peak spectra were used to identify phenolic acids, anthocyanins and their metabolites in the samples through comparison with that of authentic anthocyanidin, anthocyanin (Sigma Aldrich, Gillingham, Dorset, UK) and phenolic acid (Extrasynthese, Genay, France) standards. Standard curves (R^2^ ranges from 0.95 t0 0.99) prepared were used to quantify the identified anthocyanidins and phenolic acids in each sample. Loses due to extraction procedures were corrected with the recovery standard, 3hmPA.

### 2.12. Plasma and Urine Nitric Oxide Determination

Nitric oxide was measured as nitrite and nitrate using an HPLC-based ENO-30 NOx Analyser (Eicom Corporation, Kuoto, Japan) as described in the ENO-30 NOx analyser manual and previous studies [46,47]. ENO-30 NOx analyser is designed to perform a high sensitivity analysis of nitrite and nitrate in biological samples by combining colorimetric diazo coupling (Griess reaction) method with a reverse phase high performance liquid chromatography (HPLC). Nitrite and nitrate peaks were detected by the detector with the nitrite detection limit of 0.1 pmol at 540 nm.

The method is based on the separation of nitrate and nitrite in the sample by reverse phase HPLC followed by reduction of nitrate (NO_3_) to nitrite (NO_2_) as a result of reaction with the cadmium and reduced copper in the reduction column. The concentration of nitrite and nitrate in the sample are obtained with the aid of nitrite and nitrate standard calibration curves. 

### 2.13. Biochemical Analyses

Serum glucose, triacylglycerol (TAG), non-esterified fatty acids (NEFA), total cholesterol (TC), high density lipoprotein cholesterol (HDL-C) and C-reactive protein ultra sensitive (CRP-US) were analysed using clinical chemistry automated analyser (Instrumentation Laboratory (UK) Limited, Warrington, UK) as previously described [48]. Serum LDL-C was calculated using Friedewald equation [49]. Enzyme-based IL Test ^TM^ kits (Instrumentation Laboratory (UK) Limited, Warrington, UK) were used for the analysis of glucose, TAG, TC and HDL-C. NEFA was measured in serum using NEFA-HR (2) enzymatic colorimetric assay kit (Wako Chemicals GmbH, Nuess, Germany) as described previously [50]. CRP-US was assayed using quantex ultrasensitive kit (Biokit, Barcelona, Spain). Total antioxidant capacity (TAC) was determined using Trolox equivalent antioxidant capacity (TEAC) assay kit (Medicon, Gerakas, Greece). Plasma insulin was analysed using an enzyme-linked immunosorbent assay (ELISA)-based assay kit (Agilent Technologies Company, Dako, Denmark). Insulin sensitivity/resistance and β-cell function were estimated with the aid of a computer-based Homeostasis Model Assessment 2 (HOMA2) [51,52].

### 2.14. Statistical Analysis

All data were analysed using IBM Statistical Package for Social Sciences (SPSS) software version 21 (International Business Machines Corporation, New York, NY, USA). Data were presented as mean ± standard error in both tables and charts. Normal distribution of all datasets was checked using the Kolmogorov-Smirnov and Shapiro-Wilk, Q-Q plots and histograms. Where necessary, data were normalised, and normality confirmed using *z*-score calculated by dividing either skewness or kurtosis values by the standard error. A full factorial two-way repeated measure ANOVA was performed with treatment and time as factors. Statistical comparison of main effect (treatment) was further confirmed by Bonferroni posthoc test. Any difference that resulted in a *p*-value of less than 0.05 (95% confidence interval) was considered statistically significant. 

## 3. Results

### 3.1. Baseline Characteristics of Study Volunteers

The baseline characteristics of volunteers are summarised in Table 2. There were no significant changes in the baseline parameters between hibiscus drink and water consumption study visits. Responses of volunteers to questionnaire suggested no adverse reaction as a result of hibiscus drink consumption. 

### 3.2. Bioavailability of Hippuric Acid, Gallic Acid and Its Metabolites

Gallic acid, 3-*O*-methyl gallic acid (3OMGA), 4-*O*-methyl gallic acid (4OMGA) and hippuric acid were the main phenolic metabolites detected (Figure 2). Apart from 3OMGA, plasma concentrations for all the phenolic acids reached maximum concentration within 90 min following acute consumption of the extract of HSC. Relative to baseline, mean plasma gallic acid concentration increased by 6.94 µM (*p* = 0.017), 4OMGA increased by 3.94 µM (*p* = 0.006), 3OMGA increased by 0.15 µM (*p* = 0.09) and hippuric acid increased by 0.39 µM (*p* = 0.01). Intact anthocyanins were not detected in the plasma but their metabolite: hippuric acid reached maximum concentration at 55 ± 5 (mean ± SEM) minutes. A total of 0.6% of the ingested gallic acid was detected in plasma as 0.38% gallic acid, 0.22 4OMGA and 0.01% 3OMGA. The plasma hippuric acid detected represents 0.09% of the ingested anthocyanin.

### 3.3. Impact of Hibiscus Drink Consumption on Systolic (SBP), Diastolic (DBP) and Pulse Pressure (PP) and Heart Rate

The change in SBP, DBP, PP and heart rate up to 4 h after hibiscus drink or water consumption relative to baseline were summarised in Figure 3. There was no significant effect of treatment, time or treatment-time interaction for SBP or DBP up to 4 h post hibiscus drink consumption. Significant (*p* = 0.003) time effect was observed for the heart rate and PP (*p* = 0.011) while no significant effect of treatment or treatment-time interaction was observed. No significant difference was observed in any of the summary response measures (Table 3) when hibiscus drink was compared with water.

### 3.4. Impact of Hibiscus Drink Consumption on FMD of the Branchial Artery, Augmentation Index, Plasma and Urine Nitrate and Nitrite

The changes in percentage FMD of the brachial artery are shown in Figure 4a. There was a significant treatment (*p* = 0.017), time (*p* = 0.006) and treatment-time interaction (*p* < 0.001) for the mean FMD of the brachial artery after hibiscus compared to water consumption. Changes in the FMD of the brachial artery relative to baseline were similar with treatment (*p* < 0.001), time (*p* = 0.006) and treatment-time interaction (*p* < 0.001).

There was no significant effects observed for the changes in Augmentation Index (AIx) for the postprandial response of AIx relative to baseline. There was an overall significant (*p* < 0.001) time effect for both mean AIx and changes in mean AIx relative to baseline after the consumption of a glass of hibiscus compared to water.

Figure 4b–d respectively summarise the changes in urinary nitrite (NO_2_), nitrate (NO_3_) and NOx concentrations following hibiscus drink or water consumption relative to baseline. There was no statistical difference in the urinary NO_2_ and NO_3_ at 2 h post-consumption of hibiscus drink compared with water, yet a borderline statistical increase (*p* = 0.048) in urinary NO_3_ at 4 h post-consumption of hibiscus drink compared with water was observed. No significant effect of postprandial urinary NO_2_ response was observed. However, there was a significant (*p* < 0.001) time effect of the urinary NO_2_. Although urinary NOx increased from baseline to 4 h post hibiscus drink consumption, it decreased post water consumption, yet there was no statistical treatment (*p* = 0.05) or treatment-time interaction. There was a significant time effect (*p* < 0.001) for the urinary NOx concentration when hibiscus drink was compared with water consumption.

Figure 4e–g respectively summarise the changes in plasma nitrite (NO_2_), nitrate (NO_3_) and NOx concentration following hibiscus drink or water consumption relative to baseline. A significant treatment (*p* = 0.016) effect was observed for plasma NO_2_ up to 4 h post-consumption of hibiscus drink when compared with water, but no time (*p* = 0.190) or treatment-time interaction (*p* = 0.058) effects were observed. A significant (*p* < 0.001) time effect was observed for plasma NO_3_ and NOx up to 4 h post-consumption of hibiscus drink compared with water, but no significant treatment or treatment-time effect was observed. 

### 3.5. Impact of Hibiscus Drink Consumption on Postprandial Lipids, Biomarkers of Insulin Resistance and Inflammation

A significant time effect (*p* < 0.001) was observed with a similar postprandial pattern of response for serum TAG from 0 to 4 h post hibiscus drink or water consumption with peak concentrations (approximately 1.6 and 1.9 mmol/L increase relative to baseline after hibiscus drink or water consumption respectively) at 210 and 90 min after breakfast/baseline and lunch respectively which then subsequently declined. There was no time by treatment interaction and although lower mean postprandial TAG concentrations were observed following hibiscus drink consumption compared with water control, this did not reach statistical significance. There was no significant difference in the AUC, IAUC, C_max_ and T_max_ of mean TAG response when hibiscus drink consumption was compared with water (Table 4).

A significant time effect (*p* < 0.001) was observed for the serum NEFA response relative to baseline with a similar postprandial response which sharply declined reaching approximately 214 µmol/L less than the baseline at 90 min post hibiscus drink or water consumption. No treatment or treatment by time interactions were observed. Consumption of hibiscus drink did not cause significant difference in the postprandial summary measures of NEFA response when compared with water (Table 5).

A significant (*p* < 0.001) time effect was observed with a biphasic serum glucose response relative to baseline for both hibiscus drink and water consumption. The first phase peaked at 60 min post breakfast and the second phase peaked at 60 min post lunch (equivalent to 180 min breakfast). No significant treatment or treatment-time interaction was observed, however there was a tendency for a lower postprandial glucose response after hibiscus consumption at 30 and 210 (*p* < 0.05) minutes post consumption, but this did not reach statistical significance after Bonferroni correction (*p* < 0.006). There were no significant differences in any summary measures of postprandial glucose (Table 4).

There was a significant time effect for the postprandial insulin response which showed a biphasic response similar to the glucose response, although there was no significant treatment or time-treatment interaction. However, there was a tendency for a lower postprandial insulin response after hibiscus consumption at 30, 150 and 210 min post consumption yet this did not reach statistical significance after Bonferroni correction (*p* < 0.006). There were no significant differences in any summary measures of the postprandial insulin response (Table 4).

No significant differences were observed for any surrogate fasting measure of insulin sensitivity (HOMA2 IR, %B and %S). Assessment of glucose: Insulin ratio (GIR) showed no significant (*p* > 0.05) treatment or time-treatment interaction effect when hibiscus drink and water consumption study arms were compared. However, there was a significant (*p* < 0.001) time effect.

Acute consumption of hibiscus drink did not impact on postprandial CRP-US responses. Even though not statistically significant, the decreasing trend after HSC consumption was observed (Figure 5).

Changes in plasma total antioxidant capacity of volunteers were not significantly (*p* > 0.05) affected by treatment, time or time-treatment interaction. However, a significant increase (*p* = 0.026) in the area under the 0 to 2 h curve for the TAC response (Table 4) post hibiscus drink consumption compared to water was observed and a significantly later time to reach peak concentration (*p* = 0.007).

## 4. Discussion

In this novel human study, we were the first to determine the acute impact of HSC extract (containing 150 mg total anthocyanins and 311 mg gallic acid) consumption on BP, vascular function and other biochemical markers of CVD. As a result of acute consumption of HSC extract the postprandial SBP (primary outcome) was lower than after the water control, although this did not reach statistical significance. The use of digital automatic BP monitor and relatively small sample size could be potential limitations of our study and it is recommended that future research should consider larger sample size and the use of other acute BP measurement methods such as photoplethysmomanometry in order to record beat-to-beat BP data and short-term variability.

However, acute HSC consumption resulted in improvements in vascular function, measured by FMD of the brachial artery, and recognised as an important CVD risk marker. Our results are difficult to compare directly with other HSC extract studies as ours was the first acute postprandial study of HSC extract. However, acute consumption of anthocyanins-rich blueberries showed similar effects [38]. In a randomized, double-blind, placebo-controlled chronic study a decrease in SBP and DBP of 7.2 and 3.1 mmHg respectively resulted from consumption of HSC extract (as brewed hibiscus tea) at a daily dose of 3.75 g HSC/720 mL (containing 21.12 mg anthocyanin) for 6 weeks by pre- and mildly hypertensive adults not taking BP-lowering medication [53]. In a related study [54], SBP and DBP were decreased by 15.4 and 4.3 mmHg respectively following daily consumption of 4 g HSC/ 480 mL for 30 days by type II diabetic patients with mild hypertension. Reduction in SBP and DBP by 14.2 and 11.2 mmHg respectively was reported as a result of daily consumption of 10 g HSC/500 mL (containing 9.6 mg anthocyanin) for 4 weeks by patients with diagnosed hypertension and without antihypertensive treatment for at least 1 month [55]. Furthermore, patients with moderate essential hypertension had SBP and DBP decreases of 17.6 and 10.9 mmHg respectively following daily consumption of two spoonfuls of HSC (as sour tea)/glass of water for four weeks [56]. Therefore, it may suggest that habitual intake of hibiscus drink has a potential to reduce BP and CVD risk. If translated to a population level, the reduction in BP observed as a result of hibiscus drink consumption could give rise to reduction of CVD by up to 10% based on the assertion by Prospective Studies Collaboration [57] that suggested that a 2 mmHg usual reduction in SBP could give rise to a 10% decrease in mortality, due to stroke and 7% decrease in mortality from ischemic heart disease (IHD) and other vascular causes in middle age population.

The benefit of the consumption of the extract of HSC on BP could be because of the synergy of many nutrients. In this study we have investigated the bioavailability of anthocyanins and phenolic acids in the HSC extract following acute consumption with mixed meals in men with l-10% CVD risk. Gallic acid, 4-*O*-methylgallic acid, 3-*O*-methylgallic acid and hippuric acid were detected in the plasma following the acute intake of HSC extract. Contrary to a previous study [44] on the bioavailability of cranberry juice anthocyanins, we have not detected intact anthocyanins in the plasma, however the plasma hippuric acid detected could be a metabolite of anthocyanins contained in HSC. Our findings suggest that gallic acid and its metabolites were the most bioavailable phenolics of the HSC extract. The decline in the concentrations of 4-*O*-methylgallic acid and hippuric acid observed in this study after reaching peak plasma concentrations at 1hr relative to baseline, could be due to the metabolism, cellular uptake and excretion of the available gallic acid and anthocyanins ingested from the HSC extract.

In a related study [58], plasma hippuric acid increased by about 500% in 300 min relative to baseline after acute consumption of 400 mL of fruit and vegetable puree-based drink (FVPD) by apparently healthy volunteers. This suggested that phenolics within the fruits and vegetables were absorbed and metabolised within 3 h of consumption and were able to exert biological effects acutely. In addition, the plasma antioxidant potential, determined by the relative increase in ferric-reducing antioxidant potential (FRAP), was reported [59] to increase by 8% of baseline as a result of the acute consumption of FVPD. This supports the findings in our study, which showed a significant increase in the area under 0–2 h plasma antioxidant response curve, suggesting the potential of polyphenol-rich drinks to interact with oxidative stress signalling pathways, which could have a beneficial impact on vascular function.

A previous animal study [60] detected 17 polyphenols and metabolites including hibiscus acid, quercetin and kaempferol derivatives in the plasma of rats following acute oral ingestion (via gastric gavage) of 1200 mg/kg of HSC extract. They [60] found a plasma maximum concentration of 112.5, 6.07, 12.59, 3.77, 1.57, 2.96 and 0.81 µM of hibiscus acid, hibiscus acid hydroxyethyl ester, hydroxycitric acid, chlorogenic acid, quercetin, quercetin glucuronide and kaempferol respectively among others at ≥2 h post hibiscus extract consumption. Frank et al. [28] reported the presence of intact cyanidin and delphinidin-3-sambubiosides which reached a maximum plasma concentration of 2.23 and 1.26 ng/mL at 90 min following a single oral dose (150 mL) of the extract of HSC containing 147 mg of total anthocyanins. Variations in the plasma phenolic compounds reported could be due to differences in the calyces, extraction methods and doses. Marked inter-individual differences in plasma anthocyanins pharmacokinetics [44], the impact of the co-consumed foods or nutrients and colonic microflora [27,61] could be other reasons for the variation in bioavailability of dietary phenolic compounds.

The non-significant decrease in SBP observed in our study was consistent with the significant improvement in vascular reactivity measured by FMD and increases in the urinary excretion of nitrite/nitrate. A related study [59] has reported a 25% increase in plasma nitrite and nitrate coupled with improvement in vascular function as a result of acute consumption of fruit and vegetable puree-based drink (FVPD) by apparently healthy volunteers. However, HSC extract did not significantly affect plasma nitrite/nitrate or arterial stiffness measured by pulse wave velocity (PWA). In a previous in vitro study [62], endothelial cells pre-treated with 20 to 40 µg/mL of the extract of HSC significantly improved endothelial nitric oxide synthase (eNOS) expression and NO production. The involvement of NO and eNOS in the modulation of vascular tone and function has been well documented with NO recognised as a key vasodilator [63]. Therefore potential mechanisms for the hypotensive action of HSC drink could be multifaceted and possibly synergistic and may act through improved endothelium-dependent response [62]. Inhibition of ACE has also been suggested as a possible mechanism of the hypotensive activity of HSC drink [11,12]. 

In consideration of the possible link between BP and circulating lipids, we investigated the impact of the acute consumption of HSC extract on lipid profiles. We spend the majority of our day in a postprandial state and postprandial TAG is now recognised as an independent risk factor for CVD and therefore important to determine. However, we found no significant impact of either intervention on lipid responses. In addition, we did not observe statistically significant changes in the postprandial serum glucose, CRP, insulin and systemic antioxidant potential.

## 5. Conclusions

This study is the first to provide evidence that acute consumption of HSC extract is beneficial to vascular function through its ability to improve postprandial FMD of the brachial artery. Overall, HSC extract consumption may be a useful dietary strategy for improving postprandial vascular function and CVD risk reduction, although this requires confirmation in further studies.

## Figures and Tables

**Figure 1 nutrients-11-00341-f001:**
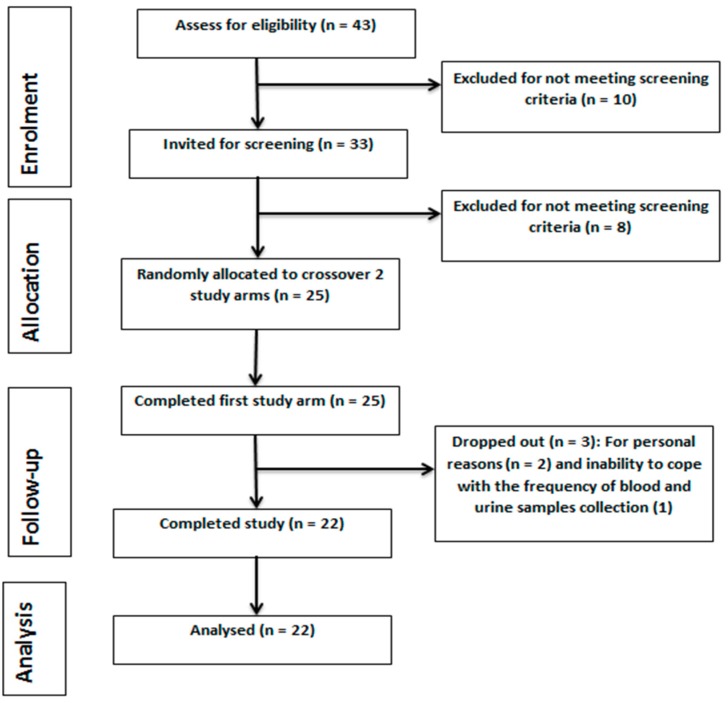
Consolidated Standards of Reporting Trials (CONSORT)-guided study flow diagram.

**Figure 2 nutrients-11-00341-f002:**
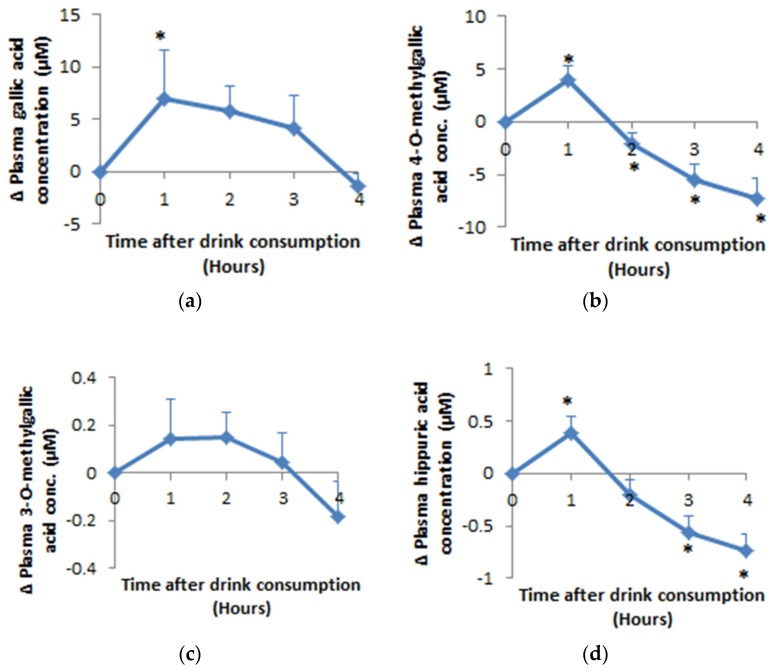
Mean (±SEM) plasma difference relative to baseline of (**a**) gallic aid, (**b**) 4-*O*-Methylgallic acid, (**c**) 3-*O*-Methylgallic acid and (**d**) hippuric acid after hibiscus drink consumption with a high fat meal followed by a medium fat meal at 2 h post drink consumption. An asterisk (*) means significant difference relative to baseline; Δ means change in.

**Figure 3 nutrients-11-00341-f003:**
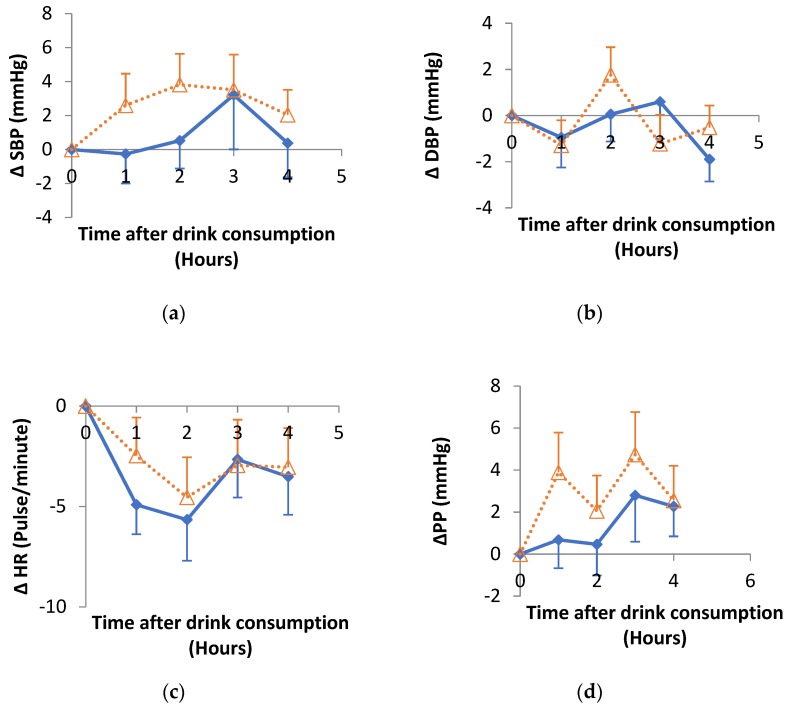
Changes in (**a**) systolic blood pressure (SBP), (**b**) diastolic blood pressure (DBP), (**c**) heart rate (HR), (**d**) pulse pressure (PP), after hibiscus drink (solid lines) or water consumption (dotted lines) relative to baseline. There were no significant differences between hibiscus and water groups in each case. Δ means change in.

**Figure 4 nutrients-11-00341-f004:**
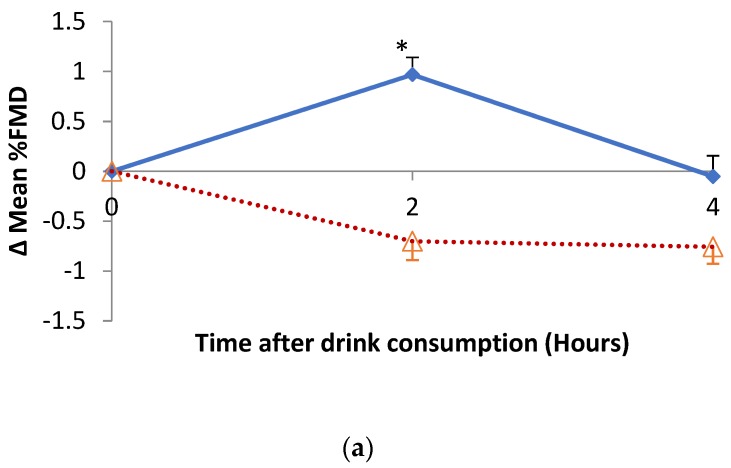
Changes in (**a**) % FMD, (**b**) urinary nitrite (NO_2_), (**c**) urinary nitrate (NO_3_), (**d**) urinary NOx, (**e**) plasma nitrite (NO_2_), (**f**) plasma nitrate (NO_3_) and (**g**) plasma NOx relative to baseline after hibiscus drink (solid lines) or water (dotted lines) consumption. An asterisk (*) means *p* < 0.001. Δ means change in.

**Figure 5 nutrients-11-00341-f005:**
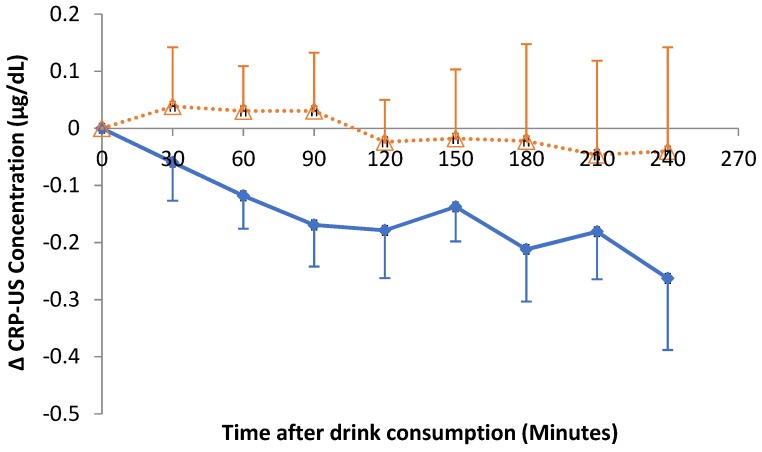
Changes in postprandial CRP-US response relative to baseline after hibiscus drink (solid line) or water (dotted line) consumption. Δ means change in.

**Table 1 nutrients-11-00341-t001:** Selected nutrient content of study breakfast and lunch.

	Breakfast	Lunch
Energy (MJ)	3.21	2.36
Fat (Saturated) (g)	50.1 (32.3)	24.9 (11.4)
Carbohydrate (g)	70.2	70.1
Protein (g)	8.6	12.8
Sodium salt (g)	1.7	1.5

**Table 2 nutrients-11-00341-t002:** Baseline characteristics of study volunteers at water and hibiscus drink study visits.

Characteristics	Water Consumption Visit	Hibiscus Consumption Visit	*p*-Values for Water vs. Hibiscus
CVD risk, age and anthropometric measures			
Age (Years)	49 ± 2.0	49 ± 2.0	1
BMI (kg/m^2^)	26.9 ± 0.7	26.7 ± 0.8	0.227
Blood pressure and vascular function measures			
SBP (mmHg)	126± 3	129 ± 3	0.063
DBP (mmHg)	74 ± 2	75 ± 2	0.225
Pulse Pressure (mmHg)	52 ± 2	54 ± 2	0.381
Heart Rate (Pulse/minute)	63 ± 3	65 ± 3	0.788
FMD (%)	3.67 ± 0.3	3.36 ± 0.2	0.838
Augmentation Index	3.45 ± 1.00	3.51 ± 0.9	0.371
Biochemical data			
TC (mmol/L)	5.35 ± 0.2	5.37 ± 0.2	0.679
HDL-C (mmol/L)	1.29 ± 0.1	1.32 ± 0.1	0.645
TC:HDL-C	4.30 ± 0.2	4.23 ± 0.2	0.881
LDL-C (mmol/L)	3.28 ± 0.2	3.32 ± 0.2	0.67
TAG (mmol/L)	1.71 ± 0.3	1.62 ± 0.2	0.901
NEFA (mmol/L)	454.9 ± 36.4	454.9 ± 38.7	0.819
Glucose (mmol/L)	5.51 ± 0.1	5.44 ± 0.1	0.843
Insulin (pmol/L)	51.90 ± 7.9	55.34 ± 9.8	0.449
HOMA2 IR	0.99 ± 0.2	1.05 ± 0.2	0.465

All values are mean ± SEM. CVD, cardiovascular disease, BMI, body mass index; SBP, systolic blood pressure; DBP, diastolic blood pressure; FMD, flow mediated dilatation; TC, total cholesterol; HDL-C, high density lipoprotein cholesterol; LDL-C, low density lipoprotein cholesterol; TAG, triacylglycerol; NEFA, non-esterified fatty acid; HOMA, Homeostasis Model Assessment; IR, insulin resistance.

**Table 3 nutrients-11-00341-t003:** Summary response measures for BP, heart rate and pulse pressure.

	SBP			DBP			Heart Rate			Pulse Pressure		
	Hibiscus	Water	*p* Value	Hibiscus	Water	*p* Value	Hibiscus	Water	*p* Value	Hibiscus	Water	*p* Value
AUC^0–240min^	31,134 ± 496	30,992 ± 571	0.426	17,924 ± 325	17,692 ± 338	0.312	14,580 ± 341	14,347 ± 374	0.324	13,210 ± 383	13,300 ± 413	0.437
IAUC^0–240min^	219 ± 361	658 ± 346	0.193	−86 ± 217	−61 ± 183	0.466	−343 ± 632	−690 ± 416	0.345	305 ± 283	719 ± 351	0.182
T_max_^0–240min^	142 ± 16	108 ± 18	0.083	101 ± 18	119 ± 17	0.237	95 ± 20	65 ± 17	0.133	138 ± 18	109 ± 18	0.130
P_max_^0–240min^	138 ± 3	135 ± 3	0.191	79 ± 1	78 ± 2	0.325	68 ± 2	66 ± 2	0.246	62 ± 2	61 ± 2	0.343
T_min_^0–240min^	95 ± 18	90 ± 19	0.428	130 ± 16	119 ± 18	0.328	91 ± 14	112 ± 17	0.179	92 ± 17	82 ± 20	0.344
P_min_^0–240min^	123 ± 2	122 ± 2	0.425	69 ± 1	69 ± 1	0.387	57 ± 1	56 ± 2	0.300	50 ± 1	49 ± 2	0.424

Values are mean ± SEM of 22 volunteers. Area under curve (AUC), incremental area under curve (IAUC), time to reach maximum pressure (T_max_), maximum pressure (P_max_), time to reach minimum pressure (T_min_) and minimum pressure (P_min_). AUC and IAUC are expressed as mmHg.minutes; T_max_ and T_min_ are in minutes while P_min_ and P_max_ are in mmHg.

**Table 4 nutrients-11-00341-t004:** Serum glucose, plasma insulin, serum TAG and plasma TAC postprandial response measures.

	Glucose			Insulin			TAG			TAC		
	Hibiscus	Water	*p* Value	Hibiscus	Water	*p* Value	Hibiscus	Water	*p* Value	Hibiscus	Water	*p* Value
AUC ^0–240 min^	1384 ± 47	1455 ± 56	0.058	49,242 ± 8666	50,017 ± 7661	0.424	547± 55	605 ± 71	0.188	343 ± 50	318 ± 40	0.105
IAUC ^0–240 min^	80 ± 40	134 ± 51	0.090	35,962 ± 6588	37,561 ± 6554	0.465	157 ± 28	196 ± 26	0.128	79.0 ± 38	75.8 ± 25	0.478
T_max_ ^0–240 min^	139 ± 14.5	135 ± 15	0.408	126 ± 13	123 ± 14	0.500	196 ± 6	196 ± 6	0.380	153 ± 14	175 ± 10	0.100
C_max_ ^0–240 min^	7.5 ± 0.3	7.6 ± 0.3	0.052	387 ± 61	403 ± 49	0.482	3.3 ± 0.3	3.6 ± 0.4	0.246	2.2 ± 0.3	2.3 ± 0.4	0.460
AUC ^0–120 min^	706 ± 23	722 ± 30	0.143	23,629 ± 3932	23,388 ± 3551	0.303	222 ± 23	324 ± 34	0.143	150 ± 17 *	130 ± 11	0.026
IAUC ^0–120 min^	53.8 ± 19.3	61.8 ± 24.6	0.209	16,988 ± 2935	17,160 ± 3100	0.383	27.5 ± 6.8	46.1 ± 19.5	0.209	18.4 ± 10.9	9.4 ± 5.3	0.209
T_max_^0–120 min^	68.4 ± 5.3	53.2 ± 5.2	0.059	64.4 ± 4.3	65.5 ± 5.1	0.191	110 ± 5	196 ± 6	0.060	96.3 ± 7.4 *	75.0 ± 8.1	0.007
C_max_^0–120 min^	6.9 ± 0.3	7.1 ± 0.3	0.201	339 ± 60	338 ± 49	0.344	2.4 ± 0.2	3.3 ± 0.3	0.201	1.7 ± 0.3	1.4 ± 0.1	0.056
AUC ^121–240 min^	678 ± 37	733± 31	0.065	25,614 ± 4838	26,629 ± 4288	0.472	239 ± 29	365 ± 43	0.065	192 ± 33	187 ± 30	0.340
IAUC ^121–240 min^	95 ± 36	109 ± 33	0.272	10,457 ± 2477	10,782 ± 2141	0.449	34.7 ± 7.9	76.3 ± 16.2	0.272	21.5 ± 16.7	50.9 ± 15.6	0.067
T_max_^121–240 min^	189 ± 7	188± 6	0.398	185 ± 5	177 ± 4	0.072	108 ± 6	196 ± 6	0.398	176 ± 8	187 ± 8	0.140
C_max_^121–240 min^	7.2 ± 0.3 *	7.5 ± 0.3	0.040	365 ± 61	347 ± 46	0.218	2.5 ± 0.3	3.6 ± 0.4	0.040	2.1 ± 0.4	2.2 ± 0.4	0.440

Values are mean ± SEM of 22 volunteers. Superscript * means significant difference when compared to water. Area under curve (AUC); incremental area under curve (IAUC); time to reach total maximum concentration (T_max_); maximum concentration (C_max_); triacylglycerol (TAG); total antioxidant capacity (TAC). AUC, IAUC, AUC and IAUC for glucose, insulin, TAG and TAC are expressed as mmol/L·minutes, pmol/L·minutes, mmol/L·minutes and mmol/L. Trolox Eq·minutes respectively. T_max_ is in minutes, C_max_ is in mmol/L for glucose and TAG; mmol/L Trolox Eq for TAC and as pmol/L for insulin.

**Table 5 nutrients-11-00341-t005:** Non-esterified fatty acids (NEFA) postprandial response measures.

	Hibiscus	Water	*p* Value
T_min_^0–90 min^	76.8 ± 4.3	76.4 ± 4.7	0.413
C_min_^0–90 min^	230 ± 16	221 ± 14	0.367
AUC ^91–240 min^	46,593 ± 3313	46,523 ± 3284	0.462
IAUC ^91–240 min^	9665 ± 2711	10,364 ± 2507	0.388
T_max_^91–240 min^	186 ± 9	195 ± 9	0.216
C_max_^91–240 min^	423 ± 26	410 ± 25	0.317
T_min_^91–240 min^	127 ± 10	130 ± 12	0.198
C_min_^91–240 min^	216 ± 15	215 ± 14	0.459

Values are mean ± SEM of 22 volunteers. Area under curve (AUC); incremental area under curve (IAUC); time to reach total maximum concentration (T_max_); time to reach total minimum concentration (T_min_); maximum concentration (C_max_) and minimum concentration (C_min_) AUC and IAUC are expressed as µM.minutes. T_max_ and T_min_ in minutes, C_max_ and C_min_ are in µmol/L.

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
