# Peer review of "Acute Effects of Hibiscus sabdariffa Calyces on Postprandial Blood Pressure, Vascular Function, Blood Lipids, Biomarkers of Insulin Resistance and Inflammation in Humans"

_nutrients, 2019, doi:10.3390/nu11020341_

Round 1
Reviewer 1 Report
The authors examined the effect of accute consuption of HSC extract on FMD, blood pressure, NO bioavailability, serum lipids, glucose and insulin as well as anti-oxidative status of serum.
The most pronounced and significant effect of the treatment was a significant but transient increase in FMD. Blood pressure and other measured parameters were not significantly affected by the treatment.
There are some minor issues that need to be addressed:
Table 1 Abbreviations should be explained in the foot note.
Fig. 3 a: labeling of the x-axis is invalid; there are some dots but (Hours) are missing.
Table 2: the first left column should be wider to properly accomodate ''min''.
lane 315: (p=0.0.016)-please correct.
lane 319: ..of hibiscus drink (were is missing) compared with water.
lane 320: ...treatment -time (effect is missing) when...
Table 3: TAC should be explained.
SUGGESTION: to highlight the major finding the title could be more focused:
Acute consumption of HSC improves endothelial function in humans
Author Response
Response to Reviewer 1 Comments
Comments and Suggestions for Authors
Point 1: The authors examined the effect of acute consumption of HSC extract on FMD, blood pressure, NO bioavailability, serum lipids, glucose and insulin as well as anti-oxidative status of serum.
The most pronounced and significant effect of the treatment was a significant but transient increase in FMD. Blood pressure and other measured parameters were not significantly affected by the treatment.
Response 1: Point 1 noted as summary of the major findings.
There are some minor issues that need to be addressed:
Point 2: Table 1 Abbreviations should be explained in the foot note.
Response 2: Abbreviations are explained in the footnote of Table 2 (previously Table 1b). Table 1 has no abbreviations
Point 3: Fig. 3 a: labeling of the x-axis is invalid; there are some dots but (Hours) are missing.
Response 3: The x-axis title in figure 3 a/c is now fully visible.
Point 4: Table 2: the first left column should be wider to properly accommodate ''min''.
Response 4: All “min” in the first column of Table 3 (previously Table 2) are properly accommodated.
Point 5: lane 315: (p=0.0.016)-please correct.
Response 5: Corrected to p=0.016. Note that the line shift because of modifications in response to comments by the reviewers.
Point 6: lane 319: ..of hibiscus drink (were is missing) compared with water.
Response 6: the missing “were” has been added. Note that the line shift because of modifications in response to comments by the reviewers.
Point 7: lane 320: ...treatment -time (effect is missing) when...
Response 7: the missing “effect” has been added. Note that the line shift because of modifications in response to comments by the reviewers.
Point 8: Table 3: TAC should be explained.
Response 8: Total antioxidant capacity (TAC) in Table 3 explained.
Point 9: SUGGESTION: to highlight the major finding the title could be more focused:
Acute consumption of HSC improves endothelial function in humans
Response 9: Suggested manuscript title would have been excellent if running titles are accepted.

Reviewer 2 Report
This study would be an interesting one, if only the authors did properly measure blood pressure, and the number of subjects wouldn't be so small.
These "acute effect" studies require other BP measurement methods, such as photoplethysmomanometry, in order to record beat-to-beat BP data and short term variability.
Although the authors calculate a sufficient statistical power for their study population, when BP is the main target other numbers of subjects are required (e.g. 5x to 10x those in the present study).
Incidentally (see Introduction, raw 43), hypertension is not "one" of the key modifiable risk factors, it is THE first and most important modifiable risk factor for CV diseases.
Author Response
Response to Reviewer 2 Comments
Comments and Suggestions for Authors
Point 1: This study would be an interesting one, if only the authors did properly measure blood pressure, and the number of subjects wouldn't be so small.
Response 1: The sample size was based on statistical power of 80%.
Point 2: These "acute effect" studies require other BP measurement methods, such as photoplethysmomanometry, in order to record beat-to-beat BP data and short term variability.
Response 2: Point 2 sounds excellent but believed it is more generic to all acute studies of this type. We shall do that in the future studies.
Point 3: Although the authors calculate a sufficient statistical power for their study population, when BP is the main target other numbers of subjects are required (e.g. 5x to 10x those in the present study).
Response 3: Point 3 is valid, but we keep it to the statistically powered sample size as it appears to be the first acute study of its type with Hibiscus sabdariffa calyces. Perhaps we may recommend subsequent studies to consider 5 to 10 times the number of subjects in the present study.
Point 4: Incidentally (see Introduction, raw 43), hypertension is not "one" of the key modifiable risk factors, it is THE first and most important modifiable risk factor for CV diseases.
Response 4: Point 4 is on raw 44 not 43. It appears that point 4 and the referred text on raw 44 are saying the same thing in different ways.

Reviewer 3 Report
1. Authors are suggested to take care of spelling and grammatical errors.
2. Please provide catalog numbers of all the kits used
3. If any pre-and post-study questionnaires has been used mention those cumulative survey points also with side effects and how authors taken care of the subjects who are allergic to hibiscus. Did they checked in the questionnaires. In inclusion criteria mentioned subjects with 1-10% of cvd risk explain exactly what it represents
4. Discussion should be provided for reason behind the depletion of 4 O methoxy gallic acid and hippuric acid in plasma.
5. Did the authors compare the compounds that were identified by HPLC in plasma with hibiscus extract which might be further beneficial? If so, please provide the table of all the compounds associated. Compounds and percentages in both extract and in plasma
6. Authors are suggested to justify how the hibiscus extract is better than the other botanical extracts and which major components associated for these beneficial effects in comparison to others.
7. Please include a paragraph in discussion how long-term consumption of extract will effect and impact
8. Pleas justify the specific reason to select only male subjects.
Author Response
Response to Reviewer 3 Comments
Comments and Suggestions for Authors
Point 1: Authors are suggested to take care of spelling and grammatical errors.
Response 1: Manuscript double checked for spelling and grammatical errors. Corrections effected were necessary.
Point 2: Please provide catalog numbers of all the kits used
Response 2: Catalog numbers added at relevant points.
Point 3: If any pre-and post-study questionnaires has been used mention those cumulative survey points also with side effects and how authors taken care of the subjects who are allergic to hibiscus. Did they check in the questionnaires? In inclusion criteria mentioned subjects with 1-10% of cvd risk explain exactly what it represents
Response 3: None of the study volunteers report any allergy, discomfort or other negative reactions to the Hibiscus drink. This information is obtained from tolerability questionnaire administered. The choice of volunteers with CVD risk of 1 to 10 % is to exclude people with high risk of CVD development, who ideally should be advised to go for medication.
Point 4: Discussion should be provided for reason behind the depletion of 4 O methoxy gallic acid and hippuric acid in plasma.
Response 4: The decline in the levels of 4-O-methylgallic acid and hippuric acid observed in this study after reaching peak plasma concentrations in 1hr relative to baseline could be due to the complete metabolism of the available gallic acid and anthocyanins ingested as components of HSC.
Point 5: Did the authors compare the compounds that were identified by HPLC in plasma with hibiscus extract which might be further beneficial? If so, please provide the table of all the compounds associated. Compounds and percentages in both extract and in plasma.
Response 5: No comparison was made because intact anthocyanins were not detected in the plasma.
Point 6: Authors are suggested to justify how the hibiscus extract is better than the other botanical extracts and which major components associated for these beneficial effects in comparison to others.
Response 6: This is captured in the first and second paragraphs of the discussion.
Point 7: Please include a paragraph in discussion how long-term consumption of extract will effect and impact
Response 7: This is covered in the first paragraph of the discussion.
Point 8: justify the specific reason to select only male subjects.
Response 8: This is because FMD, one of the secondary endpoints in this study is known to be significantly influenced by hormones. Therefore, male subjects were considered to reduce variability in FMD. However, further studies with only females and a more robust study with larger sample size should include both sexes to compare.

Round 2
Reviewer 2 Report
The response of the authors does not satisfy any of the raised points.
I do not see any improvement in the new (?) version of the manuscript.
Incidentally, I suggest that the authors review their original (first submission) manuscript: their response 4 is totally wrong.
Author Response
Response to Reviewer 2 Comments at Round 2
Comments and Suggestions for Authors
Point 1: These "acute effect" studies require other BP measurement methods, such as photoplethysmomanometry, in order to record beat-to-beat BP data and short term variability.
Response 1: Point 1 has been addressed. We have added the use of digital automatic BP monitor and relatively small sample size as potential limitations of our study. We strongly recommend that future research should consider larger sample size and use of photoplethysmomanometry in acute BP measurement in order to record beat-to-beat BP data and short-term variability.
Point 2: Incidentally (see Introduction, raw 43), hypertension is not "one" of the key modifiable risk factors, it is THE first and most important modifiable risk factor for CV diseases.
Response 2: Point two comment addressed, text revised to capture hypertension as the first and most important modifiable risk factor for CVD.
